# A 63-kDa Periplasmic Protein of the Endonuclear Symbiotic Bacterium *Holospora obtusa* Secreted to the Outside of the Bacterium during the Early Infection Process Binds Weakly to the Macronuclear DNA of the Host *Paramecium caudatum*

**DOI:** 10.3390/microorganisms11010155

**Published:** 2023-01-07

**Authors:** Masahiro Fujishima, Hideaki Kawano, Isamu Miyakawa

**Affiliations:** Department of Environmental Science and Engineering, Graduate School of Science and Engineering, Yamaguchi University, Yoshida 1677-1, Yamaguchi 753-8512, Japan

**Keywords:** *Holospora*, *Paramecium*, endosymbiont-host interactions, infection, protein secretion, DNA-binding protein, PRP1, DNA affinity chromatography, environmental adaptation

## Abstract

The Gram-negative bacterium *Holospora obtusa* is a macronucleus-specific symbiont of the ciliate *Paramecium caudatum*. It is known that an infection of this bacterium induces high level expressions of the host *hsp60* and *hsp70* genes, and the host cell acquires both heat-shock and high salt resistances. In addition, an infectious form of *H. obtusa*-specific 63-kDa periplasmic protein with a DNA-binding domain in its amino acid sequence is secreted into the host macronucleus after invasion into the macronucleus and remain within the nucleus. These facts suggest that binding of the 63-kDa protein to the host macronuclear DNA causes changes in the host gene expressions and enhances an environmental adaptability of the host cells. This 63-kDa protein was renamed as periplasmic region protein 1 (PRP1) to distinguish it from other proteins with similar molecular weights. To confirm whether PRP1 indeed binds to the host DNA, SDS-DNA PAGE and DNA affinity chromatography with calf thymus DNA and *P. caudatum* DNA were conducted and confirmed that PRP1 binds weakly to the *P. caudatum* DNA with a monoclonal antibody raised for the 63-kDa protein.

## 1. Introduction

The Gram-negative bacterium *Holospora obtusa* is a macronucleus-specific symbiont of the ciliate *Paramecium caudatum* [1,2,3,4,5,6,7]. During the life cycle of this bacterium, the infectious form (IF, about 13 μm long) of *H. obtusa* escapes from the host digestive vacuole to the cytoplasm after the bacteria are engulfed into digestive vacuoles of the host cell. The IFs invade the host macronucleus and subsequently begin to differentiate to the reproductive short form (RF; about 1.5–2 μm long). The RFs propagate by binary fissions in the host macronucleus [1,4,8,9]. When the host cell starves or the host protein synthesis is inhibited with emetine, a eukaryotic cell-specific protein synthesis inhibitor, the RFs stop dividing and differentiate into the IFs via the intermediate forms [10]. About half of the length of this IF consists of a cytoplasmic region with two nucleoids that are stained with 4′,6-diamidino-2-phenylindole dihydrochloride (DAPI). [8,11]. The other half consists of a periplasmic region involving a special tip called an invasion tip [1,8,11,12,13]. The cytoplasmic regions, periplasmic region, and invasion tip of the IF can be readily identified by a phase-contrast or a differential interference contrast microscope (DIC) [11,12]. Unlike the IF, however, the RF shows no such inner cellular differentiation [3,4,8,9,11,14].

The *H. obtusa*-bearing *P. caudatum* cells express a high level of heat shock protein genes *hsp60* and *hsp70* [15]. This allows host cells to exhibit high viability when transferred from the normal culture temperature of 25 °C to the unsuitable growth temperature of 35 °C [15]. The *H. obtusa*-free cells almost cease swimming at both 4 °C and 40 °C, although cells bearing the RF of *H. obtusa* can swim at these temperatures [16]. *Paramecium* cells bearing the micronucleus-specific *H. elegans* also express high level of *hsp70* mRNA, even at 25 °C. The host cells acquire a heat-shock resistance and survive better than symbiont-free paramecia, even at 37 °C [17]. This overexpression of the *hsp70* gene of the host cells continues irreversibly, even after removal of *H. elegans* by treatment with penicillin [17]. Nakamura et al. [18] identified six genes of *P. caudatum* aside from *hsp60* and *hsp70*, which are differentially expressed in *H. obtusa*-bearing and *H. obtusa*-free cells using differential display reverse-transcribed PCR. Furthermore, *Holospora*-bearing *Paramecium* acquires an osmotic stress resistance [19,20], and various metal chloride resistances [21]. This evidence shows that infection of *Holospora* species alters the host gene expression. However, it remains unclear how *Holospora* species affect the host gene expression. Traditionally, bacteria maintained within the ciliates were referred to as endosymbionts [22,23], though the details of their contributions to the host have not been fully elucidated. *H. obtusa*, *H. elegans*, and *H. undulata* are difficult to culture outside of the host cells due to their reduced genome size [24]. However, they contribute to their host survival by transforming the host into various stress tolerances [15,16,17,19,20,21]. On the other hand, even these *Holospora* species suppress the host cell division when they proliferate excessively in the host cells though its molecular mechanism is not yet elucidated.

Abamo et al. [25] demonstrated using a monoclonal antibody mAb3D1B9C4 raised for the 63-kDa periplasmic protein of the IF cells of *H. obtusa* that the 63-kDa proteins were secreted into the host macronucleus after bacterial invasion into the target nucleus. This 63-kDa protein is IF specific and one of the most abundant proteins of the IF. In addition, Abamo et al. [25] found that not only the pre-existing 63-kDa proteins in the IF, but newly synthesized 63-kDa proteins are also secreted after the bacterial invasion into the host macronucleus. The deduced amino acid sequence based on a cloned 63-kDa protein gene showed that this protein has a novel sequence with a putative signal peptide comprising 24 amino acids near the C-terminal of the protein. Furthermore, a Pfam motif search on the Genome Net (https://www.genome.jp/, accessed on 28 November 2022)) showed that the 63-kDa protein (periplasmic region protein 1, PRP1) has a DNA-binding domain, D5_N at aa369–491. These results strongly suggest that the 63-kDa protein might induce a change in the host gene expression by binding to the host macronuclear DNA. Therefore, this study is designed to ascertain whether the 63-kDa periplasmic protein secreted from the IF cells of *H. obtusa* can bind to the host macronuclear DNA.

## 2. Materials and Methods

### 2.1. Strains and Cultures

Symbiotic *P. caudatum* strain RB-1 cells (syngen 4, mating type E) infected with *H. obtusa* strain F1 and aposymbiotic RB-1 cells were used. Strain RB-1 was collected in 1993 in Stuttgart, Germany, by Hans-Dieter Görtz. *H. obtusa* strain F1 cells were isolated by M. Fujishima from symbiotic *P. caudatum* strain 103 collected by H.-D. Görtz (syngen and collected year and place unknown) and infected to the *P. caudatum* strain in RB-1 cells. *Paramecium* cells were cultivated in fresh lettuce juice medium with modified Dryl’s solution (MDS) [8,26] containing 0.0001% (*w*/*v*) stigmasterol (Tama Biochemical Co., Ltd., Tokyo, Japan). The MDS was inoculated with a non-pathogenic strain of *Klebsiella pneumoniae* strain 6081 one day before use at 25 °C [27]. Symbiotic and aposymbiotic *P. caudatum* cells were cultured in test tubes (18 mm × 180 mm) at 25 °C.

### 2.2. Isolation of IFs and RFs of H. obtusa

The RFs and IFs of *H. obtusa* were isolated from homogenates of macronuclei isolated from symbiotic *P. caudatum* strain RB-1 cells at the log phase and from cells at the stationary phase of growth, respectively, using Percoll density gradient centrifugation [8].

### 2.3. Indirect Immunofluorescence Microscopy

The *H. obtusa* cells isolated from the host cells were air-dried on cover glasses (4.5 mm × 24 mm), fixed with 4% (*w*/*v*) paraformaldehyde dissolved in phosphate-buffered saline (PBS) (137 mM NaCl, 2.68 mM KCl, 8.1 mM NaHPO_4_·12H_2_O, 1.47 mM KH_2_PO_4_, pH 7.2) for 10 min, permeabilized with 20 mM NaOH for 25 s to enable entry of the antibody into bacterial intracellular space [13,25], and washed three times with PBS for 3 min each. The cells on cover glasses were treated with culture supernatant of hybridoma cells containing monoclonal antibodies mAb3D1B9C4 raised for the 63-kDa periplasmic protein of the IFs [25] for 2 h at 25 °C, washed with PBS, and treated with Alexa Fluor 488 (AF488) conjugated goat anti-mouse IgG (Molecular Probes Inc., Eugene, OR, USA) diluted 1000-fold with PBS for 2 h at 25 °C. Then the cover glasses were washed with PBS, stained with 0.1 µg mL^–1^ 4′,6-diamidino-2-phenylindole dihydrochloride (DAPI), and observed under a differential interference contrast microscope (DIC) and a fluorescence microscope (BX60; Olympus Corp., Tokyo, Japan). For infection experiments, 1 × 10^7^ IFs were mixed with 1 × 10^4^
*H. obtusa*-free *P. caudatum* strain RB-1 cells for 1 h at 25 °C. The initial invasion of the IF into the host macronucleus via the host digestive vacuoles takes about 10 min, and the number of the IFs invaded in the macronucleus increases over time [28]. Then, paramecia were washed with MDS three times by centrifuging at 100× *g* for 1 min to remove bacteria that were not engulfed by the paramecia. Paramecia were suspended in MDS at 25 °C. At 1 h, 1 day, and 2 days after mixing, the cells were centrifuged, airdried on cover glass (4 mm × 24 mm) and fixed with 4% (*w*/*v*) paraformaldehyde dissolved in PBS for 10 min at room temperature. Then, the cover glasses were washed with PBS containing 0.05% (*v*/*v*) Tween 20 (PBST), treated with a culture supernatant of hybridoma cells containing monoclonal antibodies mAb3D1B9C4 for 3 h at 25 °C, washed with PBST, and treated with AF 488 conjugated goat anti-mouse IgG diluted 1000-fold with PBS for 3 h at 25 °C. Paramecia on cover glasses were then washed with PBST, stained with DAPI, and observed under a DIC and a fluorescence microscope.

### 2.4. DNA Extraction and Preparation of a DNA Affinity Column

For a DNA affinity column, DNA extracted from 5 × 10^6^ cells of the aposymbiotic *P. caudatum* strain RB-1 was used. *P. caudatum* DNA was extracted as follows. Cells of *P. caudatum* in the stationary phase of growth were washed with MDS three times by centrifuging at 300× *g* for 3 min at 4 °C; then they were suspended in DNA extraction buffer (100 mM NaCl, 10 mM Tris, 25 mM ethylenediaminetetraacetic acid (EDTA), 0.5% sodium dodecyl sulfate) before phenol–chloroform (1:1 *v*/*v*) extraction [29]. Cyanogen bromide-activated sepharose 4B (Sigma-Aldrich Corp., St. Louis, MO, USA) was swollen with 1 mM HCl. The resin was washed with deionized water (DW) and then 10 mM potassium phosphate buffer (K-PB, pH 8.0). Then the resin was suspended in 10 mM K-PB and mixed with extracted *P. caudatum* DNA. The DNA coupling resin was washed with DW to remove unreacted DNA. Then, the resin was washed with 0.2 M glycine buffer (pH 8.0) to block the active group, 10 mM K-PB, 1 M K-PB, and 1 M KCl. Finally, 200 µL of the resin was packed into a 1 mL syringe.

### 2.5. SDS-PAGE, SDS-DNA PAGE and Immunoblotting

For sodium dodecyl sulfate-polyacrylamide gel electrophoresis (SDS-PAGE) of whole cells of *H. obtusa*, 1 × 10^7^ IF cells were loaded in each lane and the gel was stained with Coomassie brilliant blue R250 (CBB) or with silver staining. For the SDS-PAGE of fractions of DNA-cellulose chromatography, each fraction was boiled with an equal volume of Laemmli’s lysis buffer [30]. Pre-stain molecular mass markers (Nacalai Tesque Inc., Kyoto, Japan) were used.

For immunoblotting, the proteins were transferred to an Immobilon-P membrane (Millipore Corp., Bedford, MA, USA). The membrane was incubated for one night at 4 °C with mAb3D1B9C4 including the culture supernatant containing 5% (*w*/*v*) skim milk as primary antibodies. Then it was incubated in alkaline phosphatase-conjugated anti-mouse IgG (Biosource International Inc., Camarillo, CA, USA) as secondary antibodies, and finally with the phosphatase substrate (KPL, Gaithersburg, MD, USA).

For SDS–DNA PAGE, a 10% (*w*/*v*) separation gel containing 10 μg mL^–1^ calf thymus DNA (Worthington Biochemical Corp., Lakewood, NJ, USA) or 10 μg mL^–1^
*P. caudatum* whole genome DNA was used. After electrophoresis, the gel was washed five times with 100 mL of TE buffer (10 mM Tris-HCl, pH 7.5, 1 mM ethylenediaminetetraacetic acid [EDTA], pH, 8.0) for 5 min each and was incubated with TE buffer containing 10 mM MgCl_2_ for one night. The gel was then stained with 1 μg mL^–1^ ethidium bromide (EB) in TE buffer for 30 min and the gel was observed under excitation by ultraviolet rays [31,32]. SDS-DNA PAGE detects nuclear histones on gels. Nuclear histones bind to DNA in gels containing DNA, and the DNA-histone complexes inhibit the binding of DNA and EB. As a result, DNA-histone complexes appear as dark bands on gels under UV illumination after EB staining of gels. An EB-staining negative band indicates the bands that were not stained with EB. SDS-DNA PAGE was also applied to detect a DNA-binding protein, Abf2p, of yeast mitochondrial nucleoids [32].

### 2.6. DNA Affinity Chromatography

For DNA affinity chromatography with calf thymus DNA, 1.0 × 10^8^ IFs of *H. obtusa* were sonicated in 200 µL column buffer (10 mM sodium–potassium phosphate buffer, Na, K-PB, pH 6.8), containing 0.1% Nonidet P-40, 1 mM EDTA and 1 mM phenylmethylsulfonyl fluoride (Sigma-Aldrich Corp.) and centrifuged at 20,000× *g* for 10 min at 4 °C. The supernatant was applied to a DNA affinity column (150 µL in the fully hydrated form) and conjugated by native calf thymus DNA (Amersham Biosciences, Amersham, UK, No. 27-5581-02). The column was washed with 1 mL 10 mM Na, K-PB, pH 6.8, containing 1 mM EDTA. Eluates from the column were collected at 100 µL each, and the fractions were numbered 1–10. Then, the column was washed again with 1 mL of 10 mM Na, K-PB containing 0.2 M NaCl and 1 mM EDTA, and the eluates were collected at 100 µL each and numbered 1′–10′. Similarly, fractionation was also conducted with 10 mM Na, K-PB containing 2 M NaCl and 1 mM EDTA, and eluates were collected at 100 µL each and numbered 1″–10″. All fractions were used for SDS-PAGE and immunoblot.

For DNA affinity chromatography with *P. caudatum* DNA, 2.0 × 10^7^ IFs of *H. obtusa* were sonicated in a 200 µL column buffer (10 mM Na,K-PB, pH 6.8), containing 0.1% Nonidet P-40, 1 mM EDTA and 1 mM phenylmethylsulfonyl fluoride; Sigma-Aldrich Corp.) and centrifuged at 20,000× *g* for 10 min at 4 °C. The supernatant was applied to a DNA affinity column. The column was washed with 4 mL wash buffer (column buffer containing 0.02 M NaCl). Eluates from the column were collected in every 200 μL. Similarly, the column was washed with 2 mL of elution buffers, which are the column buffers containing 0.1, 0.2, 0.3, and 2 M NaCl, respectively. Eluates of each column buffer with different salt concentrations were collected every 200 μL elution [33]. The supernatant of sonicated *H. obtusa* before and after application to the column and first fractions of each salt concentration were used for SDS-PAGE and immunoblotting with a culture supernatant of hybridoma cells containing monoclonal antibodies mAb3D1B9C4.

## 3. Results

### 3.1. Indirect Immunofluorescence Microscopy Using a Monoclonal Antibody against the 63-kDa Periplasmic Protein PRP1 of the Infectious form of H. obtusa

Indirect immunofluorescence microscopy with a monoclonal antibody mAb3D1B9C4 raised for the 63-kDa periplasmic protein PRP1 of *H. obtusa* infectious forms (IFs) shows that PRP1 is present only in the periplasmic region but not in the cytoplasmic region with two nucleoids of the IF (Figure 1a). Without permeabilization using 20 mM NaOH, the IFs could not be labeled with this antibody (data not shown). On the other hand, reproductive forms (RFs) were not labeled by this antibody even after the permeabilization (Figure 1b). A schematic diagram of the IF in Figure 1a is based on observations of the intracellular structures using electron microscopy [1,11,13]. The reproducibility of the results presented in Figure 1 was confirmed three times. 

Abamo et al. [25] reported that the PRP1 of the IFs was secreted into the host macronucleus at or later than 3 h after invasion of *H. obtusa* into the macronucleus. To confirm whether the secreted PRP1 in the macronucleus could be transferred to the host cytoplasm, infection experiments were conducted without permeabilization using 20 mM NaOH. One hour after mixing of the IFs with *P. caudatum* cells, an immunofluorescence of PRP1 was still not detectable in the host macronucleus, despite the presence of the IFs in the host macronucleus (Figure 2, 1 h, PRP1). Usually, the first IF invades into the target macronucleus within 10 min after mixing with *P. caudatum* cells [28], and number of the infected IFs in the macronucleus increases to around 10 cells till 1 h after mixing. The immunofluorescence of PRP1 became detectable in the macronucleus 1 day after mixing (Figure 2, 1 day, PRP1), but no fluorescence was detectable in the host cytoplasm even 2 days after mixing notwithstanding that macronuclei were showing strong fluorescence (Figure 2, 2 days, PRP1). As shown in Figure 2, the fluorescence of DAPI and PRP1 in the macronucleus on day 1 matches well. This indicates that the inability of PRP1 to translocate outside the macronucleus may be due to its binding to the macronuclear DNA or macronuclear chromatin proteins. The reproducibility of the results shown in Figure 2 was confirmed twice.

### 3.2. DNA-Binding Ability of the PRP1

The ability of PRP1 to bind DNA was tested by SDS-DNA PAGE and DNA affinity column chromatography. For both methods, calf thymus DNA and *P. caudatum* DNA were used to compare differences in their affinities to PRP1.

SDS-PAGE and immunoblotting with anti-PRP1 mAb3D1B9C4 show that the molecular weight of the antigen is about 63-kDa (Figure 3a,b) as previously conjectured from a deduced amino acid sequence of the gene [25]. SDS-DNA PAGE with calf thymus DNA shows six EB negative bands including PRP1 (Figure 3a–c), which suggests that not only PRP1 but also five other proteins of molecular weights 65, 57, 53, 40 and 36 might also have a DNA-binding ability, although the possibility that some of these bands might be DNA nucleases [31] cannot be eliminated. The reproducibility of the bands in Figure 3 was confirmed twice.

The ability of PRP1 to bind calf thymus DNA was also tested by DNA affinity column chromatography. Furthermore, whether the IF cells of *H. obtusa* have DNA-binding proteins other than PRP1 was also examined. DNA cellulose affinity column chromatography with calf thymus DNA reveals that PRP1 has DNA-binding ability because it was not uncoupled from the column by Na, K-PB, but uncoupled by Na, K-PB containing 0.2 M NaCl (Figure 4). PRP1 was not eluted from the column by subsequent elution with Na, K-PB containing 2 M NaCl (data not shown), which indicates that all PRP1 that was bound to the column was uncoupled by prior elution with Na, K-PB containing 0.2 M NaCl. Figure 4a shows an SDS-PAGE gel stained with silver and its immunoblot with mAb3D1B9C4 of fraction numbers 1–10 eluted by Na, K-PB. Figure 4b shows those of fraction numbers 1′–10′ eluted by Na, K-PB containing 0.2 M NaCl. Excess amounts of PRP1 appear in fraction numbers 1–3. The PRP1 that were able to bind the column were eluted by 0.2 M NaCl to fraction numbers 1′-5′. This result reveals that the PRP1 bound to the calf thymus DNA as was shown by SDS-DNA PAGE. At 6′ and 7′ in Figure 4b, there is no band labeled by immunoblot, but bands of approximately 63 kDa can be seen on the silver-stained gel in 6′ and 7′. Since it is known that the IF cells have a few proteins of about 63 kDa by 2D-SDS-PAGE in addition to PRP1 [25], this seems to be the reason for appearance of bands of about 63 kDa in 6′ and 7′ fractions.

Figure 4c shows high magnification of fraction number 3′ in Figure 4b. Bands of about 81, 61, 38, 35, 33, and 32 kDa appeared, which were not detected by SDS-DNA PAGE in Figure 3c. These six bands are well-matched to the CBB stained bands of the IFs in Figure 4d, indicating that these bands in Figure 4c are proteins or substances involving protein. Consequently, the results show that these six proteins might also have a DNA-binding ability or form a complex with DNA-binding proteins. However, the 57-kDa protein, as detected using SDS-DNA PAGE in Figure 3c, does not appear in Figure 4c, which suggests that the 57-kDa protein might be a DNA nuclease, as reported by Miyakawa et al. [32]. Whether these six proteins are also secreted outside the bacteria such as PRP1 or involved in the bacterial nucleoids remain unknown. Furthermore, the reason why PRP1 cannot bind DNA in nucleoids of the IFs before translocation of the proteins to the periplasmic region is also unknown. The reproducibility of bands in Figure 4 was confirmed three times.

Next, SDS-DNA PAGE and DNA affinity chromatography were conducted using DNA extracted from *P. caudatum* cells (Figure 5 and Figure 6). Figure 5a shows a CBB stained gel of *H. obtusa* IFs and Figure 5b shows its immunoblot with anti-PRP1 monoclonal antibody mAb3D1B9C4. Figure 5c shows EB negative bands. Compared with Figure 3c, many bands appeared including PRP1 in Figure 5c. The reproducibility of bands in Figure 5 was confirmed three times.

Figure 6 shows the DNA affinity chromatography with *P. caudatum* DNA. PRP1 can be detected in a supernatant of sonicated *H. obtusa* before applying it to the DNA affinity chromatography column (Figure 6a, lane 1 and Figure 6b, lane 1). After applying it to the column, PRP1 is only slightly detectable in the eluate (Figure 6a, lane 2 and Figure 6b, lane 2). This shows that PRP1 was coupled to the column. To know the strength of the binding force between PRP1 and DNA in more detail than the conditions used in Figure 4, the column was washed with a wash buffer containing 0.02 M NaCl, but the eluate by the buffer contained no PRP1 (Figure 6a, lane 3 and Figure 6b, lane 3). PRP1 bound to the column was eluted by 0.1 M NaCl elution buffer (Figure 6a, lane 4 and Figure 6b, lane 4), but it was not eluted from the column by subsequent elution buffers with 0.2, 0.3, and 2 M NaCl elution buffer (Figure 6a, lanes 5–7 and Figure 6b lanes 5–7), which indicates that all PRP1 that was bound to the column was uncoupled by the elution with 0.1 M NaCl elution buffer. This result is consistent with all PRP1 being eluted by 0.2 M NaCl elution buffer in Figure 4. As expected, this result indicates that PRP1 binds to the host DNA, but weakly. 

This DNA affinity chromatography also shows the presence of other *H. obtusa* proteins with affinity to *P. caudatum* DNA. (Figure 6a, lane 4). The reproducibility of bands in Figure 6 was confirmed three times.

To ascertain whether PRP1 binds with other *H. obtusa* proteins, co-immunoprecipitation using a fraction of 0.1 M NaCl elute that involves PRP1, mAb3D1B9C4, and protein G-beads was conducted, but co-immunoprecipitated proteins with PRP1 were not detected (data not shown). This result suggests that PRP1 interacts only with the host DNA. Bands other than PRP1 that have affinity with *P. caudatum* DNA appeared in Figure 6 lane 4 and indicate that they are not the substances that formed a complex with PRP1.

## 4. Discussion

PRP1 has a DNA-binding domain, D5_N in its amino acid sequence as shown by a Pfam motif search, and the distribution of PRP1 within the host macronucleus matches well with that of macronuclear DNA (Figure 2). The present study shows that PRP1 can bind to the calf thymus DNA and *P. caudatum* DNA. However, to confirm whether this protein is responsible for the changes in the host gene expression such as hsp60 and hsp70 detected by Nakamura et al. [18] and Hori et al. [17], the development of new technology is required to construct gene knockout strains of uncultured *H. obtusa* and is necessary for future research. Furthermore, since the DNA-binding ability of PRP1 is not so strong, it is necessary to prove that PRP1 can directly bind to DNA by experiments such as an electrophoretic mobility shift assay (EMSA). 

So far, five proteins including the 63-kDa PRP1 are known as periplasm specific proteins of the *H. obtusa* IF cell [13,25,33,34]. The 5.4-kDa, 15-kDa, 39-kDa, and 63-kDa PRP1 proteins are localized in the periplasmic region except its invasion tip [33,34]. On the other hand, the 89-kDa protein is localized exclusively inside the invasion tip, and partially exposes a part of the protein with two actin-binding motifs outside the invasion tip before invasion into the target macronucleus in the early infection process. When the IF invades the target macronucleus, the 89-kDa protein is left behind at the entry point of the nuclear envelope [13]. Only PRP1 is secreted out of the IF and remains within the host macronucleus. PRP1 is synthesized not only during differentiation from the RF to the IF, but also in an early infection process [25]. PRP1 probably acquires its DNA binding ability after translocation from the cytoplasmic region to the periplasmic region of the IF, because PRP1 could not be detected in two nucleoids in the IF as shown in Figure 1, and the PRP1 gene has a putative signal peptide for membrane penetration [25]. Co-immunoprecipitation experiments using 0.1 M NaCl elution fractions containing PRP1, mAb3D1B9C4, and protein G beads failed to detect co-immunoprecipitated proteins with PRP1 (data not shown). This suggests that PRP1 has a DNA-binding ability, but not the ability to bind to the host *Paramecium* proteins.

By two-dimensional sodium dodecyl sulfate polyacrylamide gel electrophoresis (2D-SDS-PAGE), Abamo et al. [25] reported that PRP1 is one of the major proteins of *H. obtusa* IF, and that emetine, a eukaryotic cell-specific protein synthesis inhibitor, failed to block the appearance of PRP1 in the macronuclei one day after infection. On the other hand, chloramphenicol, a prokaryotic protein synthesis inhibitor, and rifampicin, a prokaryotic RNA synthesis inhibitor, decreased the amount of PRP1 fluorescence that appear in macronuclei after infection. Abamo et al. [25] showed that PRP1 fluorescence appeared at 1 day after infection without the host protein synthesis. These results indicate that the macronuclear PRP1 fluorescence at 1 day after infection is derived from both releases of the pre-existing PRP1 and newly synthesized one [25]. Abamo et al. [25] also showed that PRP1 fluorescence in the macronucleus disappears after differentiation of the IF to the RFs, and reappeared when the RFs differentiated to the IFs by starvation of the host cells. These results suggest that *H. obtusa* IF has secretion systems to transfer the PRP1 from its cytoplasm to the periplasm, and then outside of the bacterial outer membrane. The Gram-negative bacteria are known to have six kinds of secretion systems to export proteins via two membranes, an inner and outer membrane. Type II and Type V secretion systems transfer the bacterial proteins from cytoplasm to periplasm using a Sec pathway, and then to the outside of the bacterial cell [35]. Dohra et al. [24] confirmed the presence of the Type II secretion systems in the *H. obtusa* genome. 

The Gram-negative bacteria secrete effector proteins to the extracellular space. It is known that OspF effector protein of *Shigella flexneri* regulates the host gene expression by modification of the host nuclear protein [36]. Furthermore, it is also known that the AnkA effector protein of pathogen *Anaplasma phagocytophilum* is secreted and binds to the host DNA [37]. The AnkA interacts with the transcriptional regulatory regions of the *CYBB* locus at sites where transcriptional regulators bind [38]. In the present study, DNA affinity chromatography revealed that PRP1 can bind to the host DNA. This suggests that PRP1 may also have a function in inducing changes in the host gene expressions, as shown by AnkA binding to the host DNA. Nakamura et al. [18] revealed that infection of *Holospora* alters the expressions of six genes aside from the *hsp60* and *hsp70* genes. So far, it is known that *Holospora*-infected cells have acquired osmotic stress resistance [19,20] and various metal chloride resistances [21], allowing the host to increase environmental adaptability and expand its habitat through *Holospora* infection. Research has never shown that *Holospora*-infected *Paramecium* are likely to be collected in brackish waters, but many more attempts in the field collections of *Paramecium* in brackish waters are required to confirm this possibility. However, the micronucleus-specific *H. elegans* also irreversibly induces overexpression of the host heat shock protein genes [17]. Since the transcriptional activity of the ciliate micronuclei could not be detected in the asexual reproductive phase [39,40,41], it remains unclear how *H. elegans* in the micronucleus or the micronucleus infected by *H. elegans* can change gene expressions of the macronucleus. Fujishima and Watanabe [42] proved using micromanipulation experiments that a micronucleus of *P. caudatum* has a function for morphogenesis of the cytopharinx during asexual reproduction. Furthermore, the micronucleus is suppressing the macronuclear function for the morphogenesis of the cytopharinx. Therefore, removal of the micronucleus induces an abnormally shortened cytopharinx in the amicronucleate cell. This abnormality recovers about one month after the removal of the micronucleus without transplantation of a new micronucleus, and probably the recovery of the macronuclear function by removal of the micronucleus. As expected, when a new micronucleus is transplanted into the amicronucleate cell, the length of the cytopharinx recovers immediately, even if it takes less than a month. This evidence clearly demonstrates that the micronucleus somehow can affect the macronuclear activity notwithstanding that the transcriptional activity of the micronucleus could not be detected during asexual reproduction, and the macronuclear function recovers at about one month after removal of the micronucleus. Like the unknown information exchange system between the macronucleus and the micronucleus regarding the morphogenesis of the cytopharinx, the macronucleus may be able to know the infection of the micronucleus with *H. elegans*.

On the other hand, it is known that the host *P. caudatum* with *Holospora* species in its nucleus is unable to maintain different *Holospora* species together in the same cell, although the mechanism remains unclear. For example, *H. obtusa*-bearing *P. caudatum* is unable to maintain the other *Holospora* species in the same cell. Usually, a large number of pre-existing *Holospora* species remains within the cell, and later infected *Holospora* species disappear from the cell. The preexisting *Holospora* species seem to build unknown mechanisms that prevent the proliferation of other *Holospora* species in the same cell. Thus, PRP1, which is secreted into the host macronucleus by *H. obtusa*, may indirectly play a role in inhibiting the growth of other *Holospora* species by altering the gene expression of the host macronucleus as shown in the stress resistance genes [17]. A similar phenomenon also occurs in *P. caudatum* bearing *H. elegans* (Fujishima, M. unpublished observation) or *H. undulata* [43,44]. *P. caudatum* with *H. obtusa* in the macronucleus cannot stably maintain micronucleus-specific *H. elegans* or *H. undulata* in its micronucleus. Since PRP1 does not diffuse outside the macronucleus, this phenomenon may have been caused by other *H. obtusa* substances, which diffused into neighboring micronucleus or by unknown substances produced by the host infected with *H. obtusa*.

The present study shows that PRP1 of *H. obtusa* IF has an ability to bind the host macronuclear DNA. The secretion of PRP1 in *H. obtusa* is an excellent model for studying bacterial secretion systems because secretion of PRP1 can be induced synchronously by infection of *H. obtusa* IFs, and secreted PRP1 can be detected easily by indirect immunofluorescence microscopy.

## Figures and Tables

**Figure 1 microorganisms-11-00155-f001:**
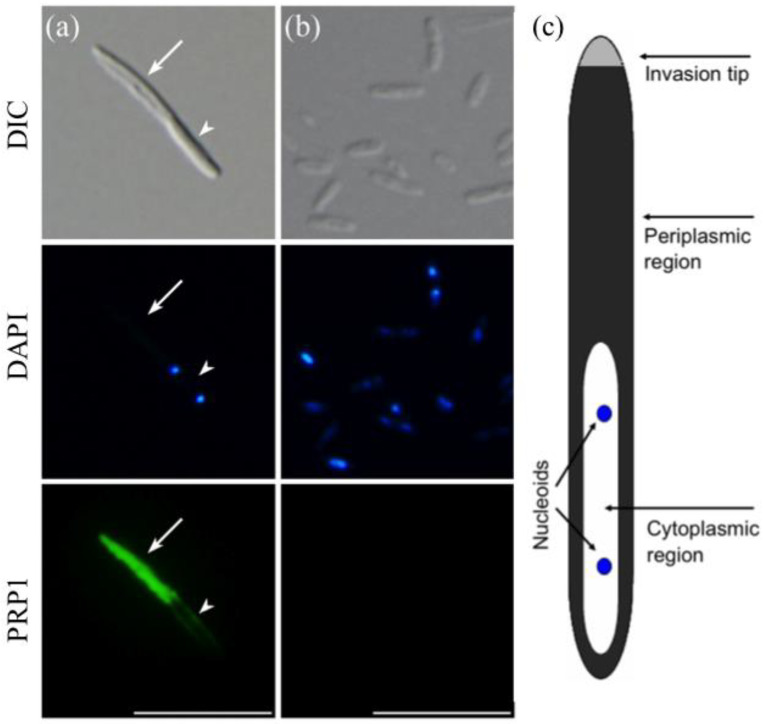
Indirect immunofluorescence micrographs of *H. obtusa* with monoclonal antibodies mAb3D1B9C4 specific for the 63-kDa periplasmic protein PRP1. Left (**a**), IF; middle (**b**), RFs; right (**c**), a schematic representation of the IF of *H. obtusa.* Upper photos, differential interference contrast (DIC); middle photos, DAPI fluorescence; lower photos, AF488 immunofluorescence. Only a periplasmic region (arrow) except an invasion tip of the IF shows immunofluorescence. Because the cytoplasmic region of the IF is covered with a thin layer of the periplasm, a faint immunofluorescence layer appears around the cytoplasmic region. A cytoplasmic region (arrowhead) with two DAPI-positive nucleoids shows no immunofluorescence. *H. obtusa* cells on cover glasses and were permeabilized with 20 mM NaOH to allow antibodies entry inside the outer membrane. The RF shows no immunofluorescence. Scale bar, 10 µm.

**Figure 2 microorganisms-11-00155-f002:**
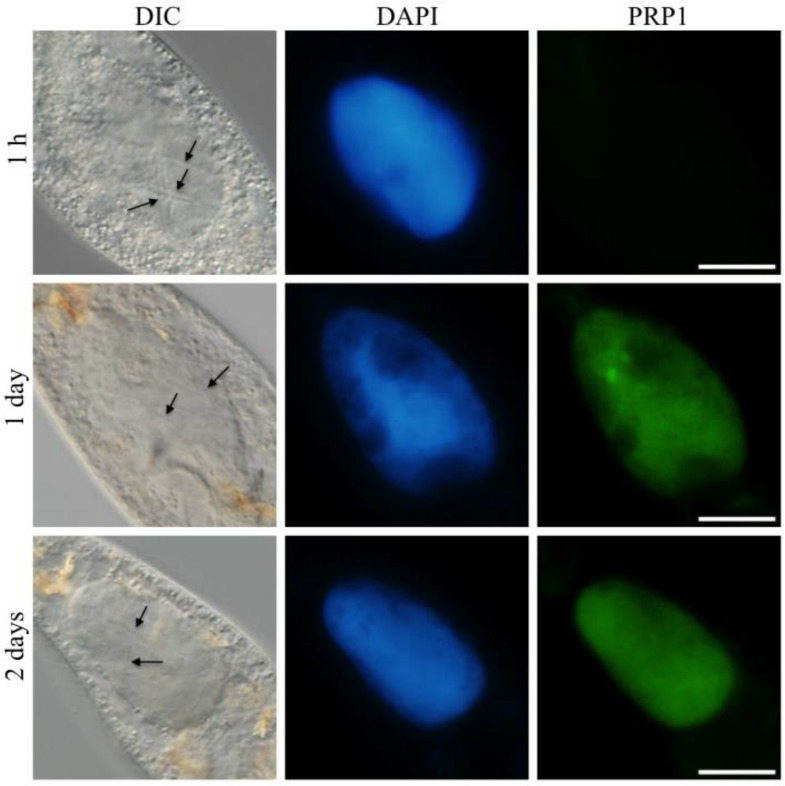
Fates of the 63-kDa PRP1 protein of *H. obtusa* in an early infection process. After mixing *P. caudatum* cells with isolated IFs of *H. obtusa*, *Paramecium* cells were fixed at 1 h, 1 day, and 2 days and labeled with an indirect immunofluorescence microscopy with monoclonal antibodies mAb3D1B9C4 specific for the 63-kDa periplasmic protein PRP1 of *H. obtusa*. *Paramecium* cells on cover glasses are not permeabilized with 20 mM NaOH, so that the PRP1 in the periplasmic region of the IFs in the host macronucleus cannot be labeled with the mAb3D1B9C4. The appearance of the immunofluorescence only in the macronucleus shows that the PRP1 secreted from the IFs cannot translocate to the host cytoplasm and is remaining in the macronucleus. Left, differential interference contrast (DIC); middle, DAPI fluorescence; right, AF488 immunofluorescence of PRP1. At 1 h after mixing with paramecia, PRP1 could not be detected in the macronucleus. At 1 day and 2 days after mixing, the macronucleus showed immunofluorescence, but the host cytoplasm did not, indicating that the PRP1 secreted from the IFs cannot pass through the macronuclear envelope. Arrows, infected IFs in the macronucleus. Scale bar, 20 µm.

**Figure 3 microorganisms-11-00155-f003:**
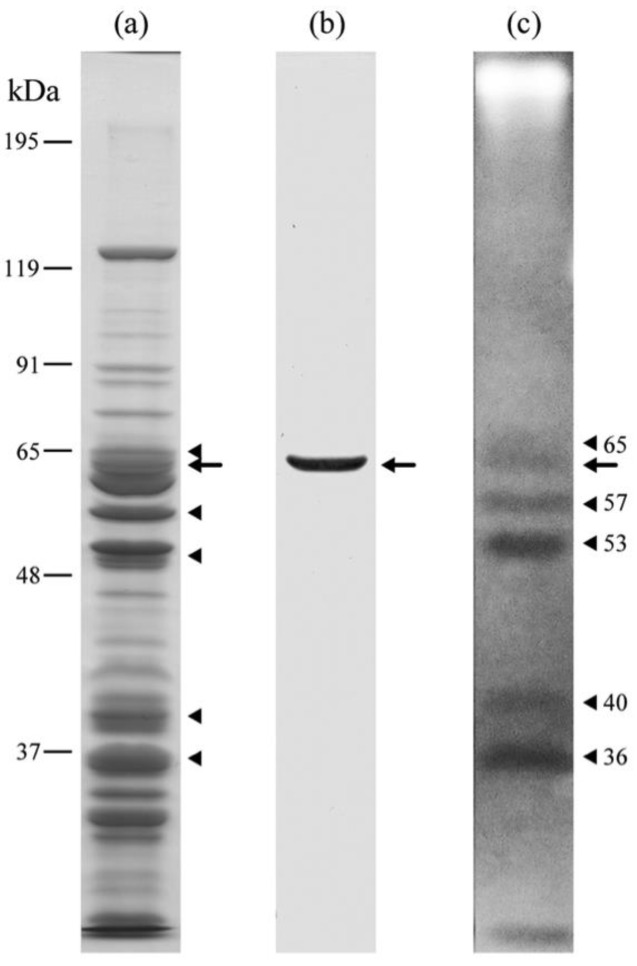
SDS-DNA PAGE of IF cells of *H. obtusa.* Calf thymus DNA was used. (**a**) CBB stained gel; (**b**) immunoblot with mAb3D1B9C4 specific for PRP1; (**c**) EB stained gel. Arrows, PRP1. Arrowheads, other EB-staining negative bands showing positions of DNA-protein complexes (see Section 2).

**Figure 4 microorganisms-11-00155-f004:**
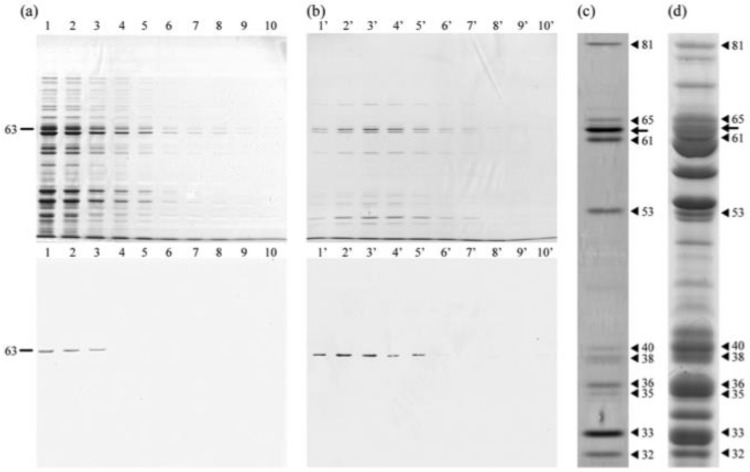
SDS-PAGE and immunoblots of fractions of DNA affinity column chromatography. Calf thymus DNA was used. (**a**) Fractions 1–10 eluted with Na, K-PB; (**b**) fractions 1′–10′ eluted with Na, K-PB containing 0.2 M NaCl. Upper, silver-stained gel. Lower, immunoblot with mAb3D1B9C4. (**c**) High magnification of a fraction number 3′ using silver staining. Not only PRP1 but also several silver staining bands were also confirmed. (**d**) CBB stained gel of IF cells of *H. obtusa*. Arrow, PRP1. Arrowheads, other bands eluted. Bands in (**c**) match the bands in (**d**).

**Figure 5 microorganisms-11-00155-f005:**
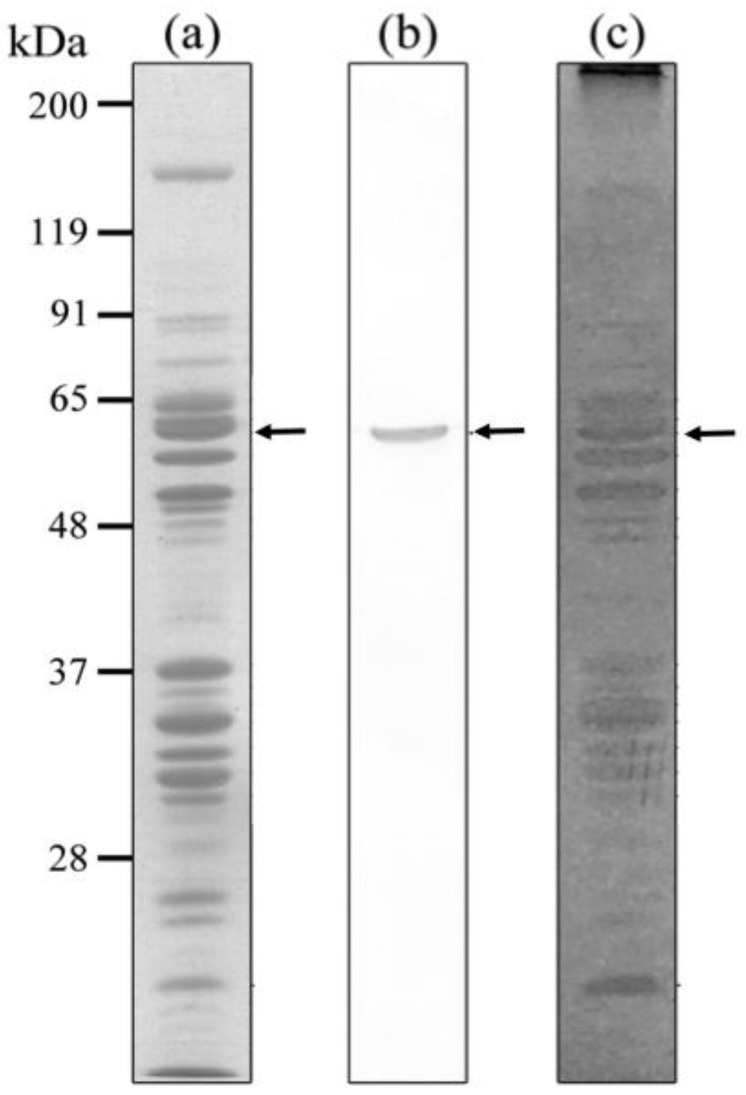
SDS-DNA PAGE of IF cells of *H. obtusa*. *P. caudatum* DNA was used. (**a**) CBB stained gel. (**b**) Immunoblot with mAb3D1B9C4 specific for PRP1. (**c**) EB stained gel. Arrows, 63-kDa PRP1. It should be noted that not only PRP1 but also many EB-negative bands that show the positions of DNA-protein complexes appeared (see Section 2).

**Figure 6 microorganisms-11-00155-f006:**
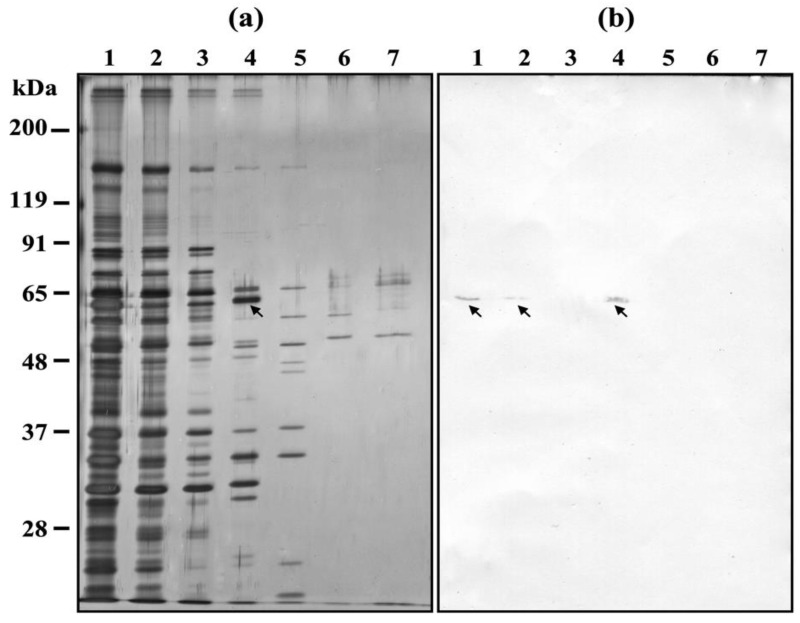
DNA affinity chromatography of *H. obtusa* IFs. *P. caudatum* DNA was used. (**a**) Silver stained SDS-PAGE gel; (**b**) immunoblotting with the monoclonal antibody mAb3D1B9C4 specific for the 63-kDa periplasmic protein PRP1 of *H. obtusa*. Lane 1, supernatant of sonicated *H. obtusa* before applying to DNA affinity chromatography column; lane 2, first elute of supernatant of sonicated *H. obtusa* after applying to the column; lane 3, first elute by elution buffer containing 0.02 M NaCl; lane 4, first eluate by 0.1 M NaCl elution buffer; lane 5, first eluate by 0.2 M NaCl elution buffer; lane 6, first eluate by 0.3 M NaCl elution buffer; lane 7, first eluate by 2 M NaCl elution buffer; arrows, PRP1. Eluates from each column were collected in every 200 μL, and the first eluate from each column was used for SDS-PAGE and immunoblot with mAb. Note that not only PRP1 but also several silver stained bands were confirmed in lane 4 of (**a**).

## Data Availability

Information on the 63-kDa protein (PRP1) gene and the whole genome sequence of H. *obtusa* can be downloaded at: https://www.ncbi.nlm.nih.gov/nuccore/AB306272.1 and https://www.ncbi.nlm.nih.gov/nuccore/NZ_AWTR00000000.2. (accessed on 28 November 2022).

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
