# Peer review of "A 63-kDa Periplasmic Protein of the Endonuclear Symbiotic Bacterium Holospora obtusa Secreted to the Outside of the Bacterium during the Early Infection Process Binds Weakly to the Macronuclear DNA of the Host Paramecium caudatum"

_microorganisms, 2023, doi:10.3390/microorganisms11010155_

Round 1
Reviewer 1 Report
I have several complaints (questions) about the discussion of the results obtained (line 289-302).
1. The claim of preferential collection of water samples with the presence of holospora's infection near estuaries. there salinity could be higher than zero, is not confirmed by any references and this statement to my knowledge is incorrect (line 289-302).
2. The lack of detectable transcriptional activity of the micronucleus does not mean its complete genetic silence (line 292-295) - see, for example, works made on the interactions of Ma and Mi in the vegetative phase of the life cycle of ciliates (Fokin, Ossipov, 1981; Fokin, 1985; Ng, 1986; Kitty et al., 1990). This set of studies was not fully completed, but they showed that the nuclear apparatus is a system of constantly interacting elements: Ma - and Micronuclei. Damage (including symbiont's invasion) or removeal of the Mi leads to a change in the spectrum of activity of the Ma )possibly because of DNA-demethylation).
3. In fact, only Holospora obtusa among of holosporas can infect the Ma of P. caudatum (line 298-303). Probably the authors mean H. caryophila, but this species of Holospora-like bacteria was recently redescribed as a representative of the new genus - Preeria (Potekhin et al., 2018), which was obvious even earlier (Fokin et al., 1996).
4. When using DIC contrast microscopy, usually the periplasmic part of the infectious form of Holospora looks brighter and more embossed than the cytoplasmic part of the cell. In Fig. 1a, the situation is the opposite, I am wandering why it could be?
5. In the preface should be emphasized the biological nature of holospora - they are rather parasites. They could be treated as symbionts only in the sense of A. de Bary definition (1878).

Author Response
Please see the atachment.

Reviewer 2 Report
Comments
In this manuscript, Fujishima et al. present that a 63-kDa periplasmic protein (PRP1) secreted from symbiotic bacterium Holospora obtusa (H. obtusa) during early infection period has the binding ability to the macronuclear DNA of the host Paramecium caudatum (P. caudatum).
They first describe the expression of PRP1 in the infectious form of H. obtusa detected by immunofluorescence with a monoclonal antibody against PRP1. Second, they focus on PRP1 expression of H. obtusa in an early infection process. Finally, they examined the DNA binding ability of PRP1 by performing the DNA affinity column chromatography.
Their findings are of great importance and would be of interest to Microorganisms' readers. However, I think there are some points that need to be done to further improve the manuscript.
Strong points: The manuscript is clearly written and all experimental procedures are well described. Procedure of DNA affinity chromatography is easy to understand, and analyses of immunofluorescence seem to be carefully performed and reliable.
Weak points:
1) Title
I am slightly concerned that the title "PRP1 binds to the macronuclear DNA of P. caudatum" is weakly supported by the results due to the reasons described below.
2) Results
The authors performed immunofluorescence to detect periplasmic expression of PRP1 in the infectious form of H. obtusa (Fig. 1). However, this reviewer is confusing because PRP1 signal indicated by an arrow in Fig. 1a seems to be cytosolic localization but not layer-like periplasmic appearance. In contrast, arrowhead is indicating weak but membrane-like fluorescence signal that is much more periplasm-like localization. How did the authors interpret these results? As one of examples of periplasmic visualization in E. coli, please refer to the paper by Ke et al. (https://journals.asm.org/doi/10.1128/JB.00864-15).
Fig. 2 should be more informative. For example, "DIC", "DAPI", and "PRP1" should be labeled in the figure. Also, "1 h", "1 day", and "2 days" should be typed on the left side of photos. Arrows and (a)-(c) should be colored in black to improve visibility. Infectious forms of H. obtusa in the macronucleus are difficult to see. Do the authors have magnified images of DIC photos?
In Fig. 3, the authors performed affinity-chromatographic analysis using DNA column with P. caudatum whole genomic DNA. Although the authors demonstrated that PRP1 protein is eluted in high salt condition (lane 4, 0.1 M NaCl) from DNA affinity column, I think that it is necessary to perform further experiments to elucidate that PRP1 directly binds to the DNA because I could not find any similar proteins, that have an ability to bind DNA, to PRP1 protein (BAF63426) via the homology-based search method like BLASTP.
At first, the authors should include positive and negative controls in this DNA affinity chromatography experiment. As the positive/negative controls, the authors are encouraged to try commercially available recombinant DNA binding proteins such as E. coli RecA and non-DNA binding proteins to prove the integrity of DNA affinity column used in this experiment. Furthermore, the authors need to perform an electrophoretic mobility shift assay (EMSA) using the purified or recombinant PRP1 protein with appropriate size of DNA probes. Without these experiments, I don't think that the authors can propose the PRP1 protein has the DNA binding ability.
3) Discussion
Does PRP1 protein specifically bind to the macronuclear DNA of P. caudatum? For example, does PRP1 protein show a binding ability to a DNA affinity column made of DNA from H. obtusa or other organisms? The authors should mention this point in discussion section.
Minor points
Fig. 3b
Do the authors have same gel photo with longer exposure because signal of all the bands is very weak?
Round 2
Reviewer 2 Report
Comments
The manuscript entitled "A 63-kDa Periplasmic Protein of the Endonuclear Symbiotic Bacterium Holospora obtusa Secreted to the Outside the Bacterium during Early Infection Process Binds Weakly to the Macronuclear DNA of the Host Paramecium caudatum." was greatly improved by this major revision. But, this reviewer still has some questions and suggestions that should be addressed by the authors before publication in Microorganisms.
In Fig. 3c&5c
The authors performed ethidium bromide staining presumably to detect the position of DNA molecules in the gel. However, what does "EB-staining negative band" mean? Does this mean "the band that was not stained with EB"? or "the negative (white) and positive (black) of the photo? This is quite confusing to me.
In Fig. 4
The authors performed DNA affinity column chromatography. But, what is difference between Fig. 4 and 6? In Fig. 4, the authors seem to use DNA cellulose affinity column, but there is no information about the column and applied sample in the figure legend. Furthermore, why did authors choose DNA cellulose column? Please briefly explain it in the result section.
In addition, "the sample" was applied and eluted from DNA cellulose affinity column under isocratic condition (Na, K-PB with 0.2M NaCl). Why didn't the authors conduct gradient or stepwise-elution as performed in Fig. 6? Under the constant salt concentration, can PRP1 bind to the column? and elute from the column under same condition? If the PRP1 protein in fractions no. 4'&5' was desalted and applied to the same DNA cellulose affinity column under 0M NaCl condition, the protein can bind to the column? and elute under 0.2M NaCl condition?
Fig. 4c
This reviewer cannot understand how the authors identified that the band with an arrow is PRP1. Did the authors perform mass spec analysis of this band? I feel that the signal intensities of the "PRP1" bands observed in silver-stained gels and immunoblots (Fig. 4a&b, upper and lower, respectively) do not match.
lanes 341-342 and 351-352
These sentences are basically same. The authors should avoid repeating same phrases.
lane 359
"molecules involving protein"
What do these molecules mean?
lane 451
"RPR1" means "PRP1"?
Author Response
Response to Reviewer 2 Comments
Point 1:The authors performed ethidium bromide staining presumably to detect the position of DNA molecules in the gel. However, what does "EB-staining negative band" mean? Does this mean "the band that was not stained with EB"? or "the negative (white) and positive (black) of the photo? This is quite confusing to me.
Response 1: The principle of EB staining in SDS-DNA PAGE and the meaning of "EB-staining negative band" were explained on Materials and Methods section (P5, L192-198). EB staining does not detect the position of DNA, but the position of DNA-protein complexes. So, EB-staining negative band means the bands that were not stained with EB.
Following sentence was added in P5, L196-197.
EB-staining negative band means the bands that were not stained with EB.
Furthermore, figure legends were rewritten as follow.
Figure legend of Figure 3
 (c) EB stained gel. Arrows, PRP1. Arrowheads, other EB-staining negative bands showing positions of DNA-protein complexes. (see Materials and Methods section).
Figure legend of Figure 4
(c) high magnification of a fraction number 3’. Silver staining. Not only PRP1 but also several silver staining bands were also confirmed.
Figure legend of Figure 5
(c) EB stained gel. Arrows, 63-kDa PRP1. It should be noted that not only PRP1 but also many EB-negative bands that show the positions of DNA-protein complexes appeared (see Materials and Methods section).
Figure legend of Figure 6
Following sentence was added.
Note that not only PRP1 but also several silver staining bands were confirmed in lane 4 of (a).
Point 2-1:The authors performed DNA affinity column chromatography. But, what is difference between Fig. 4 and 6? In Fig. 4, the authors seem to use DNA cellulose affinity column, but there is no information about the column and applied sample in the figure legend. Furthermore, why did authors choose DNA cellulose column? Please briefly explain it in the result section.
Response 2-1: In Figure 4, calf thymus DNA was used, but in Figure 6, P. cuadatum DNA was used for DNA affinity column chromatography. These explanations were added to both figure legends.
Furthermore, following explanations were added in Results section.
The ability of PRP1 to bind DNA was tested by SDS-DNA PAGE and DNA affinity column chromatography. For both methods, calf thymus DNA and P. caudatum DNA were used to compare differences in their affinities to PRP1. (P9, L303-306)
The ability of PRP1 to bind calf thymus DNA was also tested by DNA affinity column chromatography. Furthermore, whether the IF cells of H. obtusa have DNA-binding proteins other than PRP1 was also examined. (P12 L351-353)
Point 2-2:In addition, "the sample" was applied and eluted from DNA cellulose affinity column under isocratic condition (Na, K-PB with 0.2M NaCl). Why didn't the authors conduct gradient or stepwise-elution as performed in Fig. 6? Under the constant salt concentration, can PRP1 bind to the column? and elute from the column under same condition? If the PRP1 protein in fractions no. 4'&5' was desalted and applied to the same DNA cellulose affinity column under 0M NaCl condition, the protein can bind to the column? and elute under 0.2M NaCl condition?
Response 2-2: In Figure 4, the column was eluted with 0 M, 0.2 M and 2 M NaCl, while in Figure 6, with 0 M, 0.02 M, 0.1 M, 0.2 M, 0.3 M and 2.0 M NaCl. This is to examine the strength of the binding force between PRP1 and DNA in more detail in Figure 6. As the result, it was revealed that all PRP1 eluted even by 0.1 M NaCl.
If the eluted fractions containing PRP1 in Figure 4 are desalted and reapplied to the column, it should theoretically rebind to the column, and elute by 0.2 M NaCl.
Following two sentences were added to explain why 0.02 M, 0.1 M and 0.3 M NaCl elution buffers were also used in Figure 6 in addition to 0 M, 0.2 M and 2 M.
To know the strength of the binding force between PRP1 and DNA in more detail than the condition used in Figure 4, (P13, L401-403)
This result is consistent with that all PRP1 being eluted by 0.2 M NaCl elution buffer in Figure 4. As expected, this result indicates that the PRP1 binds to the host DNA, but weakly.
(P13, L411-413)
Point 3:This reviewer cannot understand how the authors identified that the band with an arrow is PRP1. Did the authors perform mass spec analysis of this band? I feel that the signal intensities of the "PRP1" bands observed in silver-stained gels and immunoblots (Fig. 4a&b, upper and lower, respectively) do not match.
Response 3:Figure 4c is high magnification of a fraction number 3’ of Figure 4b as written in the figure legend (P10, L331). Without using mass spec analysis, it appears from comparison of Figure 3a (CBB) and Figure 3b (immunoblot) that the band labeled with an arrow in Figure 4c is the PRP1. There are several major bands with similar molecular weights with PRP1 around 63 kDa as shown by 2D-SDS-PAGE of the IFs and its immunoblot (Abamo et al, 2008). This is the reason for the discrepancy in intensity between bands in Figs 4a and Fig 4b.

Following explanation sentences were added to Results section (P12, L367-372)
At 6' and 7' in Figure 4b, there is no band labeled by immunoblot, but bands of approximately 63 kDa can be seen on the silver-stained gel in 6' and 7'. Since it is known that the IF cells have a few proteins of about 63 kDa by 2D-SDS-PAGE in addition to the PRP1 [25], this seems to be the reason for appearance of bands of about 63 kDa in 6' and 7' fractions.
Point 4: These sentences are basically same. The authors should avoid repeating same phrases.
Response 4:According to the reviewer 2’s comment, sentences of P12, L368-372 in previous PDF file were removed.
Point 5:"molecules involving protein" What do these molecules mean?
Response 5:We rewrote "molecules" to "substances". (P12, L377)
Point 6:"RPR1" means "PRP1"?
Response 6:We rewrote "RPR1" to "PRP1". (P14, L473)
PS: Text written in red in the Revised manuscript is where corrections have been made including minor corrections.
